# The Preparation and Evaluation of Cyanocobalamin Mucoadhesive Sublingual Tablets

**DOI:** 10.3390/ph16101412

**Published:** 2023-10-04

**Authors:** Anwar Ma’ali, Hani Naseef, Moammal Qurt, Abdallah Damin Abukhalil, Abdullah K. Rabba, Israr Sabri

**Affiliations:** Pharmacy Department, Faculty of Pharmacy, Nursing and Health Professions, Birzeit University, Ramallah P.O. Box 14, Palestine; anwarmaali95@gmail.com (A.M.); mqurt@birzeit.edu (M.Q.); adkhalil@birzeit.edu (A.D.A.); arabba@birzeit.edu (A.K.R.); isabri@birzeit.edu (I.S.)

**Keywords:** mucoadhesive, cyanocobalamin, Permeapad^®^ permeation test, Eudragit L100-55, xanthan gum, HPC and HPMC

## Abstract

Cobalamin (vitamin B_12_), an essential vitamin with low oral bioavailability, plays a vital role in cellular functions. This research aimed to enhance the absorption of vitamin B_12_ using sublingual mucoadhesive tablets by increasing the residence time of the drug at the administration site. This research involved the preparation of different 50 mg placebo formulas using different methods. Formulas with disintegration times less than one minute and appropriate physical characteristics were incorporated into 1 mg of cyanocobalamin (S1–S20) using the direct compression method. The tablets obtained were evaluated ex vivo for residence time, and only those remaining for >15 min were included. The final formulas (S5, S8, S11, and S20) were evaluated in several ways, including pre- and post-compression, drug content, mucoadhesive strength, dissolution, and Permeapad^®^ permeation test employed in the Franz diffusion cell. After conducting the evaluation, formula S11 (Eudragit L100-55) emerged as the most favorable formulation. It exhibited a mucoadhesive residence time of 118.2 ± 2.89 min, required a detachment force of 26 ± 1 g, maintained a drug content of 99.124 ± 0.001699%, and achieved a 76.85% drug release over 22 h, fitting well with the Peppas–Sahlin kinetic model (R^2^: 0.9949). This suggests that the drug release process encompasses the Fickian and non-Fickian kinetic mechanisms. Furthermore, Eudragit L100-55 demonstrated the highest permeability, boasting a flux value of 6.387 ± 1.860 µg/h/cm^2^; over 6 h. These findings indicate that including this polymer in the formulation leads to an improved residence time, which positively impacts bioavailability.

## 1. Introduction

Vitamin B_12_ (cobalamin) is one of the essential vitamins that are not produced by the human body and is obtained by food consumption only. It plays a crucial role in human cell function and metabolism, such as the synthesis of DNA and precursors that enter the Krebs cycle, particularly in rapidly regenerating organs, such as the nervous system [1]. Human needs are 0.4 and 1.5 μg per day of vitamin B12 for infants and adults, respectively. Vitamin B_12_ deficiency (serum level is less than 200 ng/L) increases the risk of myocardial infarction and stroke and causes nervous system weakness [1,2,3].

Vitamin B_12_ is available in oral, intranasal, and parenteral dosage forms. Oral dosage forms have low bioavailability. Approximately 1–5% of free cobalamin is absorbed in the gastric system from the mouth to the intestinal mucosa [3,4,5]. While intranasal is expensive, not thoroughly researched, and unfamiliar to most individuals [6], parenterals are expensive, invasive, and require qualified healthcare personnel such as nurses [5]. Cyanocobalamin is a water-soluble compound with a molecular weight of 1355.4 Dalton (Figure 1) [7]. It belongs to the biopharmaceutical classification system (BCS) class III, with high solubility and low permeability [8], and is the most commonly used form of stable vitamin B_12_ analog [9]. The oromucosal permeation of macromolecular compounds (molecular weight > 1000 Da) is irregular and partial, resulting in inadequate absorption and reduced bioavailability [10].

The sublingual and buccal mucosa are particularly attractive for macromolecule delivery due to their thin, non-keratinized squamous stratified epithelium, which is dense and rich in vascularization [10,12,13,14,15,16]. The sublingual region is easily accessible and offers benefits such as facilitating self-management, patient adherence, and bypassing the first-pass effect [10,13]. Compared to oral administration, sublingual administration can increase drug absorption by 3–10 times, with only hypodermic shots exceeding it [15]. However, several variables may restrict drug absorption via the sublingual mucosa, including a small area for absorption, tongue movement causing shearing forces, short residence duration due to rapid turnover of saliva, involuntary swallowing of fluids greater than 200 µL, and/or involuntary swallowing of the dosage forms [10,13,17,18].

A significant factor is the length of time the formulation lasts at the absorption site (sublingual). Extending the duration of the drug’s residence time with mucoadhesive compounds that can create molecular connections with mucosal constituents, thus immobilizing dosage forms to create a longer time for drug absorption, can overcome these difficulties [12,13,17,18,19]. Owing to its low penetration resistance and negative charge, mucin significantly promotes macromolecular drug penetration into the oral mucosa via mucoadhesion [10,20]. Various mucoadhesive dosage forms, such as gel, wafers, chewing gum, and tablets, have been developed for drug delivery. When these dosage forms come into contact with saliva fluid, they become wetted and swelled on the moist mucosal surfaces, resulting in the loss of solvent and water to form a gel-like structure, thus exhibiting strong adhesion and cohesion based on the chemical nature of polymers. Chemical and/or physical interactions strengthen the adhesion [10,14,21,22]. The mucoadhesive substances and mucosal surface come into close contact due to this cohesion force, increasing the residence time of the dosage forms at the administration site [13,22,23]. Consequently, the residence time of the drug at the absorption site increases, making it resistant to the flushing effect of saliva and enhancing absorption, ultimately leading to higher plasma concentration and improved patient outcomes. [10,16,24].

This study aimed to create sublingual mucoadhesive tablets using various polymers, including xanthan gum (XG), hydroxypropyl methylcellulose (HPMC), and Eudragit. Ex vivo testing was conducted to assess the mucoadhesive strength and residence time. The drug release profile and permeability across the Permeapad^®^ membrane were determined. This research successfully resulted in sublingual tablets with a prolonged adhesion duration of over two hours, emphasizing the significance of this dosage form in enhancing bioavailability.

## 2. Results and Discussion

### 2.1. UV–Vis Spectrophotometer Analysis Method

The maximum absorbance of cyanocobalamin occurred at 361 nm. All concentrations were measured using UV–vis spectrophotometry at this wavelength. The linearity of the measurements was confirmed across a concentration of 5–40 µg/mL, with a linear regression equation of y = 0.0181x + 0.0155 and R^2^; of 0.9997. Method validation encompassed accuracy, inter-day and intra-day precision, robustness within acceptable limits, sensitivity, and stability over a three-month period (Table 1).

### 2.2. Formulation Developments without API

The formulation development procedure started with the preparation and evaluation of several formulations without cyanocobalamin. As shown in Table 10, multiple formulations have been developed to investigate the effects of different components and technologies on the tablet properties. Table 2 presents tablet weight, hardness, and disintegration time. Accordingly, successful formulas have been chosen in which the ideal disintegration time for sublingual tablets is less than three minutes [18,25]. All formulas with disintegration times less than one minute were considered successful [26]. Cyanocobalamin was then added to the successful direct compression formulas to conduct additional tests to determine the best formula for our purpose.

During the tablet preparation using the molding method, a portion of acetonitrile was evaporated, and as a result, the concentration continuously increased. Accordingly, it was difficult to estimate, control, or calculate the concentration precisely and to prepare a homogeneous solution or suspension. Because of the volatility of acetonitrile, it was continuously separated from the mixture. We note that these tablets are very fragile and crumble when handled, and their surfaces are irregular because of the evaporation process (Figure 2). As stated in a study by Rawas-Qalaji et al., mechanical properties are a challenge faced by this method, which is consistent with our results [10] (P-737).

It was observed that an increase in the concentration of mucoadhesive polymers resulted in a prolonged disintegration time (Table 2). The longer disintegration time was attributed to the increased viscosity resulting from the formation of a gel matrix on the tablet, which gradually eroded [27,28]. Furthermore, all formulations that achieved less than one minute disintegration times underwent preparation using direct compression or wet granulation methods. The direct compression method was selected as the preferred approach for completing the formulation process. It is simple, cost-effective, and requires fewer steps [29,30]. In conclusion, formulas 19, 21, 25, and 27–43 were selected for the next stage.

### 2.3. Formulation Developments with API

Based on the prior evaluations of prepared tablets without cyanocobalamin, the formulation development process was proceeded by incorporating the API into the formulations. Accordingly, numerous formulations have been developed to prepare cyanocobalamin mucoadhesive sublingual tablets (Table 11). Table 3 presents the prepared tablets’ weight, hardness, disintegration time, and residence time. 

During the residence time test, it was observed that the EC and Carb polymers (formulas S1, S2, S3, S18, and S19) exhibited rapid swelling, followed by an explosion, resulting in a residence time of less than 6 min. On the other hand, HPMC, HPC, Eudragit, and XG (S4–S17 and S20) exhibited less swelling than the previous polymers, leading to a longer residence time. This is a normal outcome since excessive swelling can create a slippery mucilage, causing easy detachment from the mucosal surface [14,31,32,33].

The residence time of the preparation increases with increasing polymer concentration, as shown in Table 3. This relationship is attributed to the characteristic behavior of the polymer. As the polymer concentration increased, more polymer chains penetrated the mucosal surface, leading to substantial mucoadhesive properties. This increase in mucoadhesion strength follows a certain trend until a critical concentration is reached [23]. Importantly, among these formulas, S5, S8, S11, and S20 demonstrated significantly longer residence times, exceeding 15 min. These formulas can be considered successful formulations owing to their extended residence times.

### 2.4. Evaluation of Mucoadhesive Sublingual Tablets 

Based on the values presented in Table 4, the blend exhibited excellent flow characteristics, as indicated by the angle of repose. Additionally, S5, S8, and S20 demonstrated good flow properties, while S11 showed a fair flowability characteristic, as determined by Carr’s index and Hauser ratio. This is related to the presence of a high percentage of microcrystalline cellulose, which exhibits excellent flow properties [34].

The physical evaluations of the tablets are presented in Table 5. The weight variation of all formulations was within the range of 49.935 ± 0.668- 50.53 ± 0.591 mg. The diameter and thickness were similar for all formulations except for S5, which contained HPMC polymers and had lower thickness. The average hardness of the formulations ranges from 4.28 to 4.94 kilopascal (KP), with all tablets exhibiting hardness values between 3 and 7. The friability percentage was <1%, ranging from 0.262 ± 0.060 to 0.480 ± 0.078. These results indicate that all formulations exhibit good mechanical properties, making them suitable for mechanical shipping and storage [26,35]. The assay confirmed the content uniformity, with values ranging from 93.508 ± 0.001 to 103.910 ± 0.004.

The surface pH test was conducted to assess the potential in vivo side effects associated with alkaline and acidic pH values, which may cause mucosal irritation. The target pH range was determined to be 6.2–7.6 in normal saliva, indicating a nearly neutral pH [19,24,36]. The surface pH of the tablets fell between 5.350 ± 0.026 and 6.630 ± 0.010, with most tablets maintaining a relatively stable surface pH, except for S11, which showed a slight decrease in pH related to acidic properties of EL100-55 polymer [34,37]. None of the formulas caused irritation, as observed in vitamin B_12_ buccal mucoadhesive films with a pH of 5.1 that did not exhibit any irritation [38].

All formulations exhibited mucoadhesive strengths within the range of 11 ± 1 to 18.670 ± 1.528 g, representing the force required to detach the tablets from the mucosal layers. The mucoadhesive strength can be arranged in ascending order as follows: S8 < S5 < S20 < S11. Additionally, S11 containing EL100-55 demonstrated the longest mucoadhesive time, lasting 118.2 min in a previous result Table 3.

As observed in Table 3 and Table 5, the S5 formula, containing HPMC, exhibited superior mucoadhesive characteristics in terms of strength and time compared to the S8 formula, which contained HPC polymer. The differences in mucoadhesive characteristics can be attributed to the swelling and viscosity of polymers [39]. HPMC has a more complex structure than HPC, potentially contributing to superior mucoadhesive properties due to its higher viscosity, facilitating stronger interactions with the mucosal surface [39,40,41]. The presence of hydroxypropyl and methoxyl groups alone increases the polymer’s hydrophilicity with a hydrophobic group. This structure allows for the formation of hydrophobic interactions in addition to fast wetting and spreading that promote the entanglement of polymer chains upon contact with the mucin surface [40]. Additionally, HPMC possesses the ability to hold fluid within its structure through pores, forming a hydrogel that enhances its ability to form hydrogen bonds with mucin, the major component of mucus. These interactions promoted adhesion and prolonged the residence time of the formulation on the mucosal surface [35,42,43]. 

The S20 formula (XG) demonstrates superior mucoadhesive characteristics compared to the S8 formula (HPC). The mucoadhesive properties of XG are primarily attributed to its charge, ionization, higher molecular weight, and wetting properties [10,21,44]. XG is a water-soluble hydrophilic polymer that rapidly dissolves in hot and cold water, allowing the polymer chains to quickly diffuse into the mucosal surface. This fast diffusion and wetting process facilitates the creation of a strong interaction between the matrix and mucosa [19,44]. Additionally, the anionic nature of XG enhances its electrostatic interactions with mucin, making it a more potent mucoadhesive than HPC, a natural non-ionic polymer [10,21,28,45]. The high molecular weight of XG (2 × 10^6^–20 × 10^6^ Dalton) plays a crucial role in its mucoadhesive behavior. Research studies have indicated that polymers with molecular weights exceeding 100,000 generally demonstrate improved mucoadhesive properties. With molecular weights in the range mentioned above, XG exhibits an exceptional capacity for mucoadhesion [23,44].

The S11 formula, containing EL100-55, exhibits the best mucoadhesive characteristics. EL100-55 is an anionic, hydrophobic, and soluble polymer derived from acrylic and methacrylic acid. It demonstrates solubility at PH levels above 5.5, such as in saliva [32,37,46]. Due to its anionic nature and charged properties, EL100-55 generates a stronger electrostatic interaction in comparison to natural non-ionic cellulose derivative polymers [21]. Its mucoadhesive characteristics can be attributed to the presence of carboxylic acid groups in the polymer, which allows it to form strong hydrogen bonds that contribute to the adhesion to the mucin in the mucosa layer. Furthermore, the high molecular weight of EL100-55 with long polymer chains promotes entanglement within the mucus layer, increasing the overall adhesion between the polymer and mucosal surface [23,31].

### 2.5. Drug Release Test

Polysaccharide polymers, such as HPMC, EL100-55, XG, and HPC, are commonly employed to control drug release from polymer matrices. These polymers facilitate drug release through a dissolution process involving solvent diffusion and/or disentanglement of polymer chains [47].

A comparison of the drug release profiles between the standard cyanocobalamin and the final formulas revealed that the standard cyanocobalamin exhibited detectable absorption after 15 min, whereas in samples S5, S8, and S11, the absorption was delayed until 1 h (Figure 3 and Figure 4). In the case of sample S20, the burst effect of the XG polymer resulted in the drug release being detected after 30 min. Therefore, the presence of the polymer in the tablets effectively retards the release of the drug from the tablet matrix [28,44]. Throughout the duration of this study (up to 23 h), the release of standard cyanocobalamin remained higher than the release of all samples. However, both the samples and the standard exhibited release percentages lower than 82.75% within this timeframe. This may be attributed to the loss of force that was responsible for transferring the drug from the donor to the acceptor compartment. Additionally, during the stability test conducted for standard cyanocobalamin in SSF at 37 °C, which mimicked the conditions of the drug release test, a loss of approximately 5.34% was observed. 

Based on the results of the drug release mathematical kinetic models for the first 5.5 h (Table 6, Figure 5), the Peppas–Sahlin model exhibits the best fit to the kinetic release data, with R^2^ values exceeding 0.99. Evidently, the drug release mechanisms in the final formulas involve Fickian and non-Fickian diffusion. The presence of mucoadhesive polymers plays a significant role in these release mechanisms.

According to the Peppas–Sahlin model results, the K_1_ constant represents the contribution of Fickian diffusion to drug release, whereas the K_2_ constant represents the contribution of non-Fickian (super Case II transport) release mechanisms, which are associated with polymer swelling, chain relaxation, and erosion. 

As observed in Table 6, drug release in all final formulas is controlled by the involvement of both the Fickian and non-Fickian release mechanism. In the case of the S5 formula (5% HPMC), it was observed that K_2_ has a higher value than K_1_ (which was negative), indicating the dominance of super Case II transport (polymer relaxation-swelling, chain relaxation, or erosion) during drug diffusion release [48]. When the HPMC polymer matrix came into contact with the dissolution medium, the solvent diffused into the matrix, leading to the swelling of the polymer and hydration, resulting in the formation of a viscous gel. The drug release is controlled by diffusion, where the drug diffuses through the swollen polymer matrix following Fick’s law. Additionally, it was associated with more complex release patterns, including relaxation with the slow erosion of the HPMC polymer matrix. The presence of solid bridges formed between polymers and drugs supports sustained drug release over time [49,50].

For S8 (15% HPC), the hydrophilic nature of the polymer results in rapid hydration and swelling. The drug release from the polymer depends on pore formation and the erosion rate of the polymer, which is influenced by the concentration of the polymer and the resulting viscosity. The more viscous hydrophilic polymer leads to slower swelling and resistance to erosion processes, thereby retarding drug release. This behavior was observed in the drug release profile (Figure 3) [49,50]. 

For S11 (15% EL 100-55), hydrophobic polymers (polymethacrylates) generate hydrogels that entrap the drug within them. When the hydrogels form, the drug is released from the adhesive polymers based on diffusion within the polymer chains, in accordance with Fick’s law. Concurrently, the entrapped drug is slowly released through polymer erosion and degradation, which follows a non-Fickian drug release pattern [49,51,52].

S20 (0.5% XG) demonstrates Fickian diffusion with non-Fickian Case II transport. This is attributed to the swelling with the initial burst drug release effect and relaxation of XG, along with the drug diffusion through the hydrophilic polymeric matrix. As the concentration increases, the viscosity also increases, leading to retarded drug release [53]. Moreover, XG and EL 100-55 have higher drug release than cellulose derivative polymers (HPMC and HPC) related to higher swelling properties, so more surface area is available for drug release [21,44,53]. This behavior is supported by Figure 3 and Figure 6. 

Based on the results obtained using the DD solver program within the initial 5.5 h for all formulations, it was observed that all formulations adhered to the Peppas–Sahlin model, exhibiting the highest R^2^ values compared to alternative models. Consequently, the Peppas–Sahlin model equation was employed to accurately depict the fitting of these models, allowing us to predict the drug release behavior during this specific time frame. Figure 6 shows the drug release profiles over the initial 5.5 h, representing the percentage of drug release. The calculated R^2^ values from the fit line were as follows: HPMC = 0.9993, HPC = 0.9988, Eudragit L100-55 = 0.9974, and Xanthan = 0.9982. A higher R^2^ value obtained from the fitted curve signifies a superior fit of the data and a stronger correlation between the Peppas–Sahlin model and our experimental findings, elucidating the drug release process’s involvement of both Fickian and non-Fickian mechanisms [54]. 

### 2.6. PermeaPad^®^ Permeation Result

The Permeapad^®^ membrane is an artificial biomimetic membrane commonly employed to investigate drug permeation from dosage forms. It is particularly relevant for studying the permeability of drugs through mucosal surfaces such as the buccal and gastrointestinal mucosa. The Permeapad^®^ membrane is designed with two supported hydrophilic sheets, and within it is a phospholipid layer “ sandwich structure,” which is formed using soy phosphatidylcholine (PC) S-100 [55,56]. The Permeapad^®^ membrane mimics the lipid composition and structure of biological membranes. This unique characteristic makes it highly suitable for reliably assessing passive drug permeation behavior and evaluating drug delivery systems [57].

As shown in Figure 7, the permeation of cyanocobalamin from S5 (HPMC) and S8 (HPC) was lower than that from S11 (EL100-55) and S20 (XG). This was consistent with the drug release profile shown in Figure 4. The observed difference in permeation can be attributed to the varying amounts of drug available in the donor compartment for permeation through the Permeapad^®^ membrane.

By referring to the drug release profile in Figure 4, it can be observed that S20 (XG) initially exhibits slightly higher drug release behavior during the first 5 h, after which the drug release profile becomes similar to that of S11 (EL100-55). This difference in the early drug release behavior could be attributed to a burst effect associated with the higher swelling of XG than that of EL100-55 [53]. While the permeation test showed that S11 (EL100-55) had higher permeability than S20 (XG), this difference in drug permeability behavior can be attributed to several factors.

First, the higher viscosity of XG compared to that of EL100-55 may play a role; EL100-55 generally has a lower viscosity. The XG viscosity of KG tends to increase at higher pH values and in the presence of salts, such as NaCl or KCl, at elevated temperatures. On the other hand, EL100-55 exhibits solubility at pH values higher than 5.5. Changes in pH can affect the solubility and ionization of polymers [44,56]. Additionally, XG has more hydrophilic groups; for the drug to cross this membrane, it should possess a balance of lipophilic and hydrophilic properties that mimic the Permeapad^®^ membrane.

In contrast, EL100-55 is an amphiphilic methacrylic acid polymer, which contains both hydrophilic and hydrophobic groups with a higher affinity for cyanocobalamin (a weak base). These characteristics promote the migration of the soluble polymer from the SSF (pH 6.8) through the Permeapad^®^ membrane to PBS (pH 7.4) along with cyanocobalamin. Therefore, EL100-55 exhibits a better balance of these properties than XG, which may enhance the interaction and increase the permeability of cyanocobalamin from the EL100-55 matrix [37,44,56,58,59,60].

The bioavailability of S11 (Eudragit L100-55), estimated by the cumulative amount of drug passing through the Permeapad^®^ membrane into the donor compartment, is approximately 12.03%. When comparing this bioavailability with that of conventional oral dosage forms (1 mg) by passive diffusion, which typically has a bioavailability of 1.3%, it is evident that the cyanocobalamin mucoadhesive sublingual tablets formula in S11 is a promising approach to significantly increasing the bioavailability of cyanocobalamin and enhancing its therapeutic efficacy [61,62].

Table 7 presents the R^2^ values for cyanocobalamin permeability, steady-state flux, and apparent permeability coefficient (*P_app_*) through the Permeapad^®^ membrane. Notably, the highest values for all parameters were observed for the S11 formula (EL 100-55). This indicates that S11 exhibits the highest cyanocobalamin permeability among the tested formulations. Specifically, the *P_app_* value of cyanocobalamin for S11 is approximately two-fold higher compared to S5, 1.6-fold higher for S8, and 1.38-fold higher for S20.

The *P_app_* values for all formulas are higher than 1.5 × 10^−6^, with a standard deviation of less than 27%. These values indicate good permeability as they exceed the cut-off value for classifying permeability [63]. However, it is important to note that the results of all drug permeation parameters consistently demonstrated that the permeation of S11 (EL100-55) was higher than that of S20 (XG) in the tested formulas. This conclusion is supported by the findings presented in Figure 7 and Table 7, which show higher permeation levels for S11 than for S20.

### 2.7. Drug Stability Test in Simulated Saliva Fluid

According to the *British Pharmacopoeia*, the acceptable range for the drug content of cyanocobalamin in tablets is 90–115% [64]. Based on the observations presented in Table 8, the maximum amount of drug lost after 24 h under conditions mimicking the administration site was 672.195 µg (5.34%). This indicates that the drug loss remains within acceptable limits, with 94.66% remaining of the initial drug amount. Considering these findings, it is recommended that 110% of the desired cyanocobalamin amount be initially added to the tablet formulation. This ensures that, even after the expected drug loss, the remaining amount will not fall below 100% (1 mg/tablet), which is the desired target. By accounting for the anticipated drug loss, the formulation could be optimized to maintain the desired drug content throughout the shelf life of the tablet.

As shown in Table 9 below, the drug was stable after three months of storage with an RSD of less than 2.

## 3. Materials and Methods

### 3.1. Materials 

Cyanocobalamin was donated by Planet Pharma, Eudragit (S100, L100, and L100-55) by Evonik Industries, and Mannitol by Pharmacare PLC. Hydroxy propyl cellulose (M.W. 100,000) and hydroxy propyl methyl cellulose ((2% aq. Soln., 20 °C) 7500–14,000 mPa.s) were purchased from Alfa Aesar (Haverhill, MA, USA); ethyl cellulose, microcrystalline cellulose PH 101, polyplasedone, and magnesium stearate from Colorcon^®^ (Milan, Italy); sublingual bovine mucosa from a local butcher (Palestine); PermeaPad^®^ membrane from innoME GmbH (Espelkamp, Germany); dialysis tubing cellulose membrane and polyvinyl pyrrolidine (MW 40,000) were purchased from Sigma-Aldrich (St. Louis, MO, USA). Xanthan gum, carbopol, sodium hydroxide pellets, sodium chloride, disodium hydrogen phosphate, potassium dihydrogen phosphate, potassium chloride, absolute anhydrous ethanol 100%, acetonitrile, and hydrochloric acid 37% were obtained from Birzeit University laboratories (Ramallah, Palestine).

### 3.2. UV–Vis Spectrophotometer Analysis Method

Cyanocobalamin analysis was performed using a UV–visible double-beam spectrophotometer from PerkinElmer (Woodbridge, ON, Canada). The analytical method was validated according to ICH Q2B guidelines, including limit of detection (LOD), limit of quantification (LOQ), accuracy, linearity, and precision [65]. 

### 3.3. Formulation of Mucoadhesive Sublingual Tablets

Initially, a variety of formulations were developed using different methods without the addition of active ingredients. The strengths of the materials used, such as the diluents, binders, disintegrants, and polymers, were modified. The excipients utilized in the current research were chosen based on the findings of a compatibility study performed one month before use in order to confirm that there is no interaction with the API, and based on the results, we selected an excipient with an assay of 99% or higher after one month. Some excipients (such as microcrystalline cellulose, magnesium stearate, PVP, and HPMC) were chosen due to the fact that they have previously been used in commercial products. Three methods were used to prepare the suggested sublingual tablet formulations: direct compression, wet granulation, and molding. In the direct compression, the excipients were mixed with a mortar and pestle for 5 min before being compressed by a manual single-punch tablet compression machine. For the wet granulation, the excipients (including 50% diluents, disintegrant, and the entire amount of binder) were mixed with a mortar and pestle, followed by gradually adding 100% ethanol until granules were formed. The granules were then sieved through mesh #16 and allowed to dry before compression. Finally, for the molding, diluent, disintegrant, and binder were dispersed in an acetonitrile solvent until a homogenous suspension was formed. A portion of the produced suspension was poured into empty medication strips. After 24 h, the tablets were completely dried and removed from the mold [10,26]. Table 10 presents the quantity of each material per tablet as a percentage, where the total weight of each tablet is 50 mg. Finally, many formulations were chosen, each with an optimal disintegration time of less than one minute.

**Table 10 pharmaceuticals-16-01412-t010:** The composition of mucoadhesive sublingual tablet formulations without cyanocobalamin.

Materials	PVP	XG	Carb	Es100	EL 100	EL 100-55	HPC	EC	HPMC	MAN	MCC	PolyP	Mg.S	Method
Formula														
1	1%	-	-	-	-	-	-	-	-	-	70	29	-	Molding
2	1%	-	-	-	-	-	-	-	-	-	70%	29%	-	Wet
3	-	-	-	-	-	-	-	-	1%	-	70%	29%	-	Wet
4	3%	5%	-	-	-	-	-	-	-	-	86.5%	5%	0.5%	DC
5	3%	5%	-	-	-	-	-	-	-	-	83.5%	8%	0.5%	DC
6	3%	5%	-	-	-	-	-	-	-	-	76.5%	15%	0.5%	DC
7	0%	5%	-	-	-	-	-	-	-	-	79.5%	15%	0.5%	DC
8	0%	5%	-	-	-	-	-	-	-	-	74.5%	20%	0.5%	DC
9	0%	3%	-	-	-	-	-	-	-	-	76.5%	20%	0.5%	DC
10	0%	3%	-	-	-	-	-	-	-	-	66.5%	30%	0.5%	DC
11	3%	5%	-	-	-	-	-	-	-	86.5%	-	5%	0.5%	DC
12	0%	5%	-	-	-	-	-	-	-	79.5%	-	15%	0.5%	DC
13	1%	5%	-	-	-	-	-	-	-	-	67%	27%	-	Wet
14	1%	3%	-	-	-	-	-	-	-	-	69%	28%	-	Wet
15	1%	4%	-	-	-	-	-	-	-	10%	79.5%	5%	0.5%	DC
16	1%	4%	-	-	-	-	-	-	-	25%	64.5%	5%	0.5%	DC
17	1%	2%	-	-	-	-	-	-	-	-	70%	27%	-	DC
18	-	1%	-	-	-	-	-	-	-	-	70%	29%	-	DC
19	1%	-	-	-	-	-	-	-	1%	-	70%	28%	-	DC
20	1%	-	-	-	-	-	-	-	5%	-	67%	27%	-	Wet
21	1%	-	-	-	-	-	-	-	5%	-	67%	27%	-	DC
22	1%	-	-	-	-	-	-	-	10%	-	61%	28%	-	DC
23	1%	-	-	-	-	-	-	-	15%	-	56%	28%	-	DC
24	-	0.5%	-	-	-	-	-	-	-	-	70%	29.5%	-	Wet
25	-	0.5%	-	-	-	-	-	-	-	-	70%	28.5%	-	DC
26	1%	-	5%	-	-	-	-	-	-	-	67%	27%	-	DC
27	-	-	0.5%	-	-	-	-	-		-	70%	29.5%	-	DC
28	-	-	1%	-	-	-	-	-	-	-	70%	29%	-	DC
29	1%	-	-	5%	-	-	-	-	-	-	66%	28%	-	DC
30	1%	-	-	10%	-	-	-	-	-	-	61%	28%	-	DC
31	1%	-	-	15%	-	-	-	-	-	-	56%	28%	-	DC
32	1%	-	-	-	5%	-	-	-	-	-	66%	28%	-	DC
33	1%	-	-	-	10%	-	-	-	-	-	61%	28%	-	DC
34	1%	-	-	-	15%	-	-	-	-	-	56%	28%	-	DC
35	1%	-	-	-	-	5%	-	-	-	-	66%	28%	-	DC
36	1%	-	-	-	-	10%	-	-	-	-	61%	28%	-	DC
37	1%	-	-	-	-	15%	-	-	-	-	56%	28%	-	DC
38	1%	-	-	-	-	-	5%	-	-	-	66%	28%	-	DC
39	1%	-	-	-	-	-	10%	-	-	-	61%	28%	-	DC
40	1%	-	-	-	-	-	15%	-	-	-	56%	28%	-	DC
41	1%	-	-	-	-	-	-	5%	-	-	66%	28%	-	DC
42	1%	-	-	-	-	-	-	10%	-	-	61%	28%	-	DC
43	1%	-	-	-	-	-	-	15%	-	-	56%	28%	-	DC

PVP: polyvinyl pyrollidene; XG: xanthan gum; Carb: Carbopol 940; Es 100: Eudragit S100; EL 100: Eudragit L100; EL100-55: Eudragit L100-55; HPC: hydroxypropyl cellulose; EC: ethyl cellulose; HPMC: hydroxypropyl methylcellulose; MAN: mannitol; MCC: microcrystalline cellulose; PolyP: polyplasidone; Mg.S: magnesium stearate.

Based on the assessment of the prepared tablets without active ingredients (Table 10), the formulation development process was followed by incorporating the API into the successfully prepared formulations, and several formulations were prepared (Table 11). Because of the very low amount of active pharmaceutical ingredients (API) (2%, 1 mg) provided to the formula, the addition of powder materials during the mixing process was performed by geometric mixing in all formulas produced using the direct compression method [19]. All the processes were performed under dim light.

**Table 11 pharmaceuticals-16-01412-t011:** The composition of suggested cyanocobalamin mucoadhesive sublingual 50mg tablet formulas.

Formula #	B_12_	PVP	EC	HPMC	HPC	EL100-55	EL100	ES100	XAN	CR940	MCC	POLY	MG.S
S1	1	0.5	2.5	-	-	-	-	-	-	-	31.75	14	0.25
S2	1	0.5	5	-	-	-	-	-	-	-	29.25	14	0.25
S3	1	0.5	7.5	-	-	-	-	-	-	-	26.75	14	0.25
S4	1	0.5	-	0.5	-	-	-	-	-	-	33.75	14	0.25
S5	1	0.5	-	2.5	-	-	-	-	-	-	31.75	14	0.25
S6	1	0.5	-	-	2.5	-	-	-	-	-	31.75	14	0.25
S7	1	0.5	-	-	5	-	-	-	-	-	29.25	14	0.25
S8	1	0.5	-	-	7.5	-	-	-	-	-	26.75	14	0.25
S9	1	0.5	-	-	-	2.5	-	-	-	-	31.75	14	0.25
S10	1	0.5	-	-	-	5	-	-	-	-	29.25	14	0.25
S11	1	0.5	-	-	-	7.5	-	-	-	-	26.75	14	0.25
S12	1	0.5	-	-	-	-	2.5	-	-	-	31.75	14	0.25
S13	1	0.5	-	-	-	-	5	-	-	-	29.25	14	0.25
S14	1	0.5	-	-	-	-	7.5	-	-	-	26.75	14	0.25
S15	1	0.5	-	-	-	-	-	2.5	-	-	31.75	14	0.25
S16	1	0.5	-	-	-	-	-	5	-	-	29.25	14	0.25
S17	1	0.5	-	-	-	-	-	7.5	-	-	26.75	14	0.25
S18	1	-	-	-	-	-	-	-	-	0.25	33.75	14.75	0.25
S19	1	-	-	-	-	-	-	-	-	0.5	33.75	14.5	0.25
S20	1	-	-	-	-	-	-	-	0.25	-	33.75	14.75	0.25

### 3.4. Evaluation of Cyanocobalamin Sublingual Mucoadhesive Blend

The final mixture was evaluated in terms of various parameters, including the angle of repose, tap, and bulk density. Carr’s index and Hauser ratio were calculated to assess the flowability characteristics of the powder. Ultimately, the powder underwent compression. The flow characteristics of the powder were assessed by analyzing the angle of repose, Carr’s index, and Hauser ratio based on USP <1174> criteria [66,67].

### 3.5. Evaluation of Sublingual Mucoadhesive Tablets

Weight, hardness, and disintegration time of the prepared placebo sublingual mucoadhesive tablets were evaluated. The disintegration test was performed by adding the compressed tablets to a beaker containing 100 mL of distilled water (DW) at 37 °C, stirring somewhat to simulate the disintegration tester [68], then turning on the timer and recording the disintegration time for each formula. While cyanocobalamin sublingual mucoadhesive tablets were evaluated for several quality control tests, weight variation was performed on a random selection of 20 tablets, the thickness and diameter of the tablets were measured using a caliper, and the hardness of 10 randomly selected tablets was assessed using a tablet hardness tester (Pharma Test Apparatebau AG, Hainburg, Germany). The means and standard deviations (SD) of the measurements were subsequently calculated. Additionally, 6.5 g of tablets were placed in the friability drum tester (Pharma Test Apparatebau AG, Hainburg, Germany) and rotated at 25 rpm for 4 min, where weight loss should not exceed 1% of the tablet’s initial weight. With no signs of cracks, capping, or breakage [18,25,46,69].

### 3.6. Content Uniformity

Ten tablets were crushed from each formula, and an amount equivalent to 1 mg of cyanocobalamin was added to 50 mL of simulated salivary fluid (SSF) at PH 6.8. The mixture was subjected to sonication for 10 min (bath sonicator from Elmasonic (Singen, Germany)), filtered, and the concentration was determined by measuring the absorption using a UV–vis spectrophotometer at 361 nm [19,27].

### 3.7. Surface PH

Six tablets were immersed in 20 mL of phosphate buffer-adjusted distilled water with a pH of 6.8 and kept in a water bath at 37 °C for 2 h (water bath shaker from Mrc laboratory instruments, London, UK). Subsequently, the pH was measured using a digital pH meter once the reading reached a constant value [19,24,36].

### 3.8. Ex Vivo Mucoadhesive Residence Time (RT)

The mucoadhesive RT was performed on excised sublingual mucosa from bovines [69], where the mucosa was cut into appropriately small pieces (1 × 3 cm) and washed. The sublingual mucosa was fixed to a plastic slide with superglue adhesive and attached to the paddle of the dissolution test using a plastic rubber (Appendix A). The tablet was then wetted with approximately 200 µL of DW and gently pressed for 30 s on the excised tissue. The slide was then immersed in dissolution vessels (USP II dissolution apparatus, Pharma Test Apparatebau AG, Germany) containing 900 mL of DW at 37 °C with 50 rpm rotation. The time at which a tablet either detached or disintegrated from the mucosal surface was considered the mucoadhesive residence time (*n* = 3) [14,16,25,35].

The formulations that passed were chosen based on their hardness, disintegration, and residence time results, where the disintegration time is less than 1 min, which was adopted previously, and the residence time for the formula is more than 10 min. After that, all formulas that succeeded were scaled up from 50 to 400 tablets.

### 3.9. Mucoadhesive Strength (MS)

For the MS testing, a specially designed model resembling a balance was utilized. One bovine sublingual mucosa was fixed to a wooden square attached to the floor of the balance model, while the other mucosa was secured to a plastic cup using a thread. A wet tablet of SSF was placed between the two mucosal surfaces and gently compressed for 30 s (Appendix A). Additional plastic cups were placed on the opposite side of the thread. A weight scale was used to gradually increase the force until the tablet detached from the mucosa (*n* = 3) [16,19].

### 3.10. Drug Release Test

The drug release test was performed using a modulated system comprising a fixed plastic tube with a 2.5 cm diameter immersed in a 100 mL glass beaker containing 50 mL of SSF at pH 6.8 and agitated using a magnetic stirrer (multi-stirrer and stir bar were obtained from VELP Scientifica, Usmate Velate, Italy). The lower open end of the tube was covered with a cellulose dialysis membrane secured by plastic rubber, which was pre-wetted in SSF buffer at pH 6.8 for 30 min. One tablet and 5 mL of SSF were added to the interior of the tube [38]. The beaker was placed in a water bath at 37 °C and maintained using two pumps that circulated water from the bath into the glass dish and vice versa (Appendix A). The test was conducted for both the standard and final four formulas. The time intervals for the standards were 0, 0.25, 0.5, 0.75, 1, 1.5, 2, 3, 5, and 23 h, while for the samples, they were 1, 1.5, 2.5, 3.5, 4.5, 5.5, and 22.5 h; then, a 10 mL sample was collected and replaced with a fresh SSF solution (*n* = 3). The cumulative drug amount was determined by measuring the absorption using a UV–vis spectrophotometer at 361 nm. The obtained values were plotted against time to create a graph, and various mathematical kinetics models were fitted to the data to determine the drug release kinetic using the DD solver program. A higher R^2^ value indicates a better fit and provides insight into the release mechanism of cyanocobalamin from sublingual mucoadhesive tablets. 

### 3.11. PermeaPad^®^ Permeation Test

Drug permeation tests were performed using a Franz diffusion cell (ORCHID ScientificTM, Nashik, India) equipped with a Permeapad^®^ membrane with an exposed area of 20 mm. The experiment involved two compartments. In the donor compartment, 2 mL of the samples was added to one tablet of SSF at PH 6.8, mimicking the conditions of the oral mucosa. The acceptor compartment was filled with 20 mL of phosphate-buffered saline (PBS) at pH 7.4 and 37 ± 0.5 °C with 250 rpm stirring (Appendix A). These buffer solutions represent the physiological conditions under which the drug is released and permeates the membrane (*n* = 3) [17,19,55]. At each hour interval, 1 mL of samples were collected from the acceptor compartment and replaced with fresh PBS. The collected sample was then diluted with 2 mL of PBS and measured at 361 nm using a UV–vis spectrophotometer.

The cumulative drug amount, steady-state flux (*J*), and apparent permeability coefficient (*P_app_*) were calculated for each sample using Equations (1) and (2), respectively. The calculation of *P_app_* allows for a quantitative assessment of permeability, which is valuable in understanding the drug’s ability to cross the sublingual mucosa and potentially reach the systemic circulation [55,56,57].
(1)$$J=\frac{dn\;}{\left(A\times dt\right)}$$
where *J* is the steady state flux (µg/min), *dn* is the cumulative amount of permeated drug (µg), *dt* is time (min), and *A* is the area of permeability (cm^2^) [56].
(2)$$P_{app}=\frac{J}{C_{0}}$$
where *P_app_* is the apparent permeability coefficient (cm/s), *J* is steady-state flux of the cyanocobalamin through the membrane, and *C*_0_ is the initial concentration of cyanocobalamin in the donor compartment (µg/mL).

### 3.12. Drug Stability Test in Simulated Saliva Fluid

For stability assessment, 10 mg of cyanocobalamin was added to 50 mL of SSF in a 500 mL volumetric flask, and the volume was adjusted to the mark; the resulting concentration is 20 µg/mL. The solution was then incubated in a water bath at 37 °C for 24 h, and the samples were collected at 0, 1, 2, 3, 4, 5, 6, and 24 h. The absorbance was measured at 361 nm using a UV–vis spectrophotometer, and the drug concentration, recovery %, and drug amount in each sample were determined.

Additionally, cyanocobalamin stability in solution was assessed by preparing a stock solution (1000 µg/mL). This involved dissolving 500 mg of cyanocobalamin in an appropriate volume of SSF solution in a volumetric flask and adjusting the total volume to 500 mL using SSF at PH 6.8. The mixture was then sonicated for 10 min to ensure complete dissolution. Subsequently, a sample from the stock solution was diluted to a concentration of 20 µg/mL with SSF, and its absorption was measured. Additionally, the stock solution was stored at room temperature for one week and for 90 days in a closed brown volumetric flask containing SSF solution. After these storage periods, the absorption was measured again at a concentration of 20 µg/mL.

## 4. Conclusions

This study offers valuable insights into the preparation and evaluation of cyanocobalamin mucoadhesive sublingual tablets using various polymers. The formulated tablets achieved an optimal residence time with an appropriate drug release mechanism using appropriate techniques and analytical methods. These results have important implications for the development of optimized formulations for sublingual drug delivery, potentially enhancing the bioavailability and the therapeutic efficacy of cyanocobalamin. 

Among the tested formulas, four formulas exhibited strong mucoadhesive characteristics with the sublingual mucosa containing xanthan gum, HPMC, HPC, and Eudragit L100-55 polymers. These formulas remain to adhere for a minimum of 22 min and require a force of up to 11 g for tablet detachment. Among these four formulas, S11 (containing Eudragit L100-55) emerged as having the best mucoadhesive properties; S11 exhibited an impressive attachment time of 118.2 min and a detachment force of 26 ± 1 g. It also exhibited a favorable drug release profile, releasing up to 76.85% over a 22 h period, allowing for the efficient permeation of most cyanocobalamin molecules with an apparent permeability coefficient reaching 2.188 × 10^−6^ cm/s. 

These results show that the choice of polymer greatly affects mucoadhesive properties, drug permeation, and drug release, suggesting that it could be used to prepare more bioavailable dosage forms.

## Figures and Tables

**Figure 1 pharmaceuticals-16-01412-f001:**
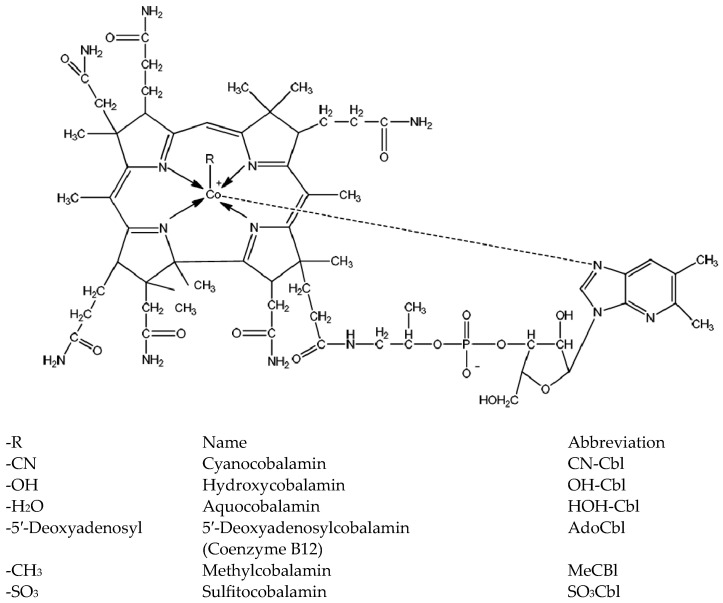
Analogs of vitamin B12 structures [11].

**Figure 2 pharmaceuticals-16-01412-f002:**
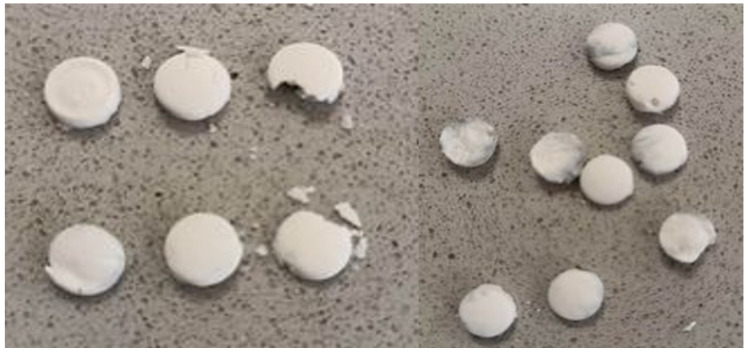
Mucoadhesive sublingual tablets after 24 h by molding method.

**Figure 3 pharmaceuticals-16-01412-f003:**
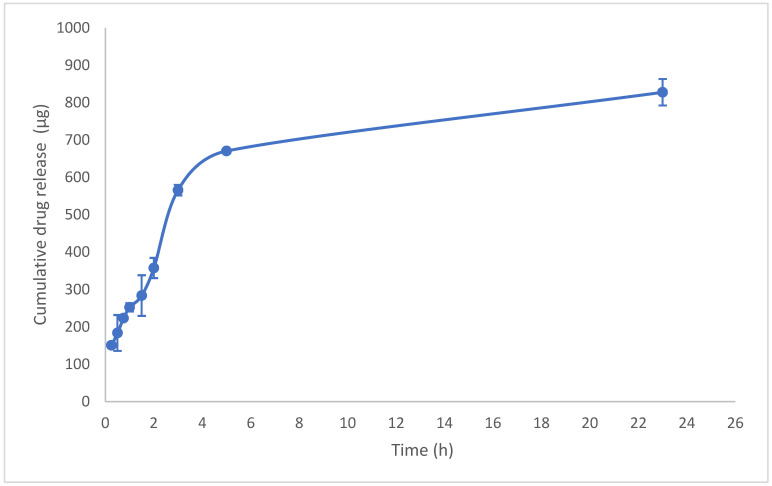
The result of standard cyanocobalamin cumulative drug release.

**Figure 4 pharmaceuticals-16-01412-f004:**
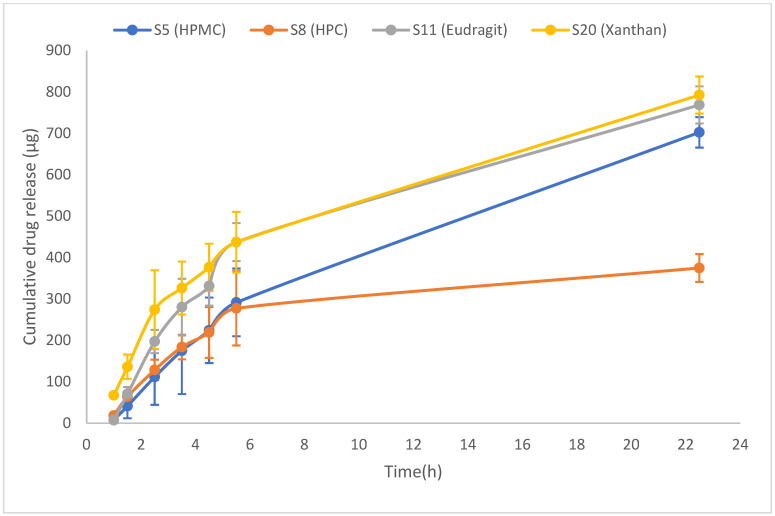
The result of the cumulative drug release test for final formulas S5, S8, S11, and S20 (HPMC, HPC, EL100-55, and XG).

**Figure 5 pharmaceuticals-16-01412-f005:**
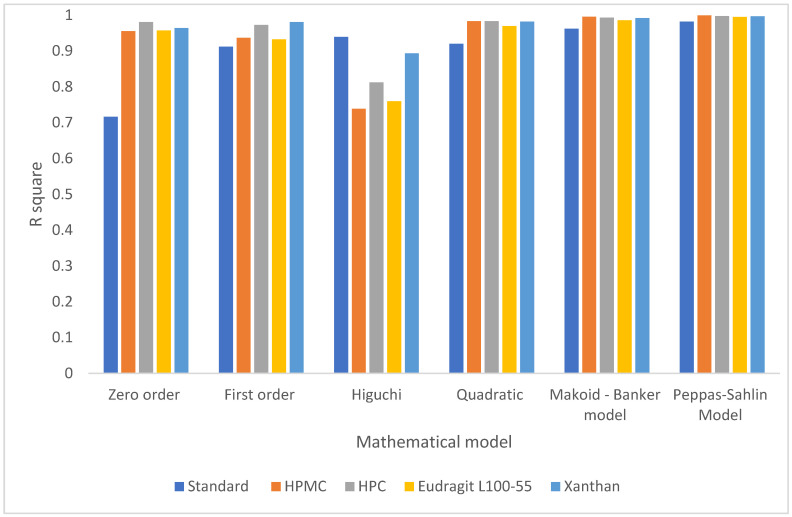
A comparison of R^2^ values for the various kinetic mathematical models.

**Figure 6 pharmaceuticals-16-01412-f006:**
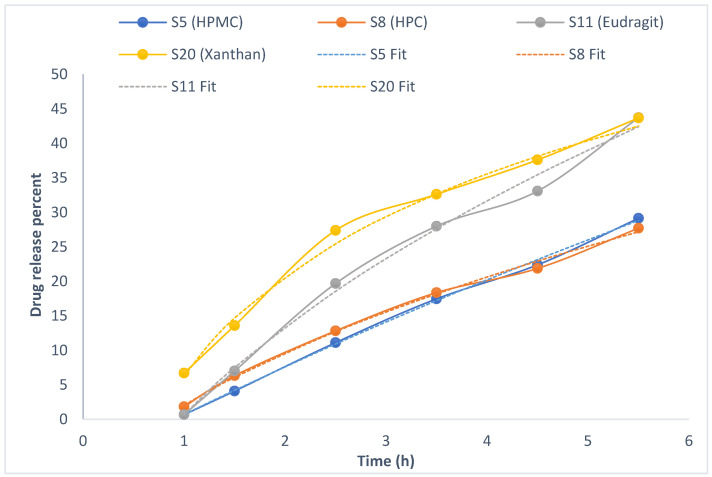
In vitro drug release profiles with a fitting line up to 5.5 h in the Peppas–Sahlin model calculated and indicated by the value of R^2^.

**Figure 7 pharmaceuticals-16-01412-f007:**
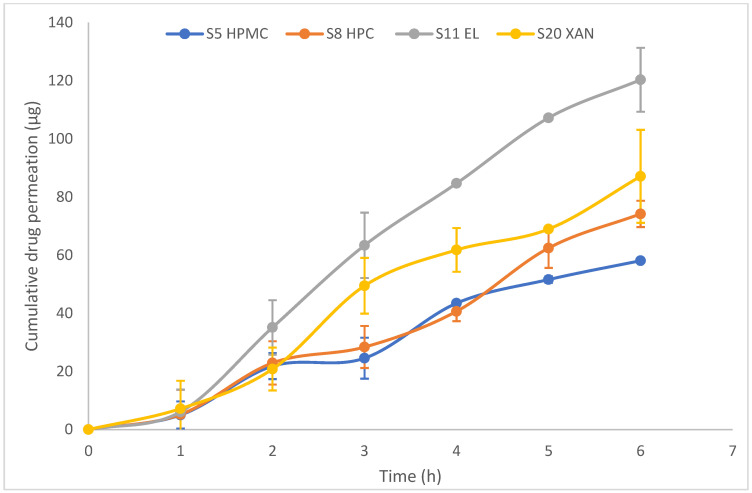
The result of Permeapad^®^ cyanocobalamin permeation test.

**Table 1 pharmaceuticals-16-01412-t001:** The result of cyanocobalamin analysis method.

Parameter		Result	Assay	RSD	Accepted Criteria
Calibration curve range		5–40 µg/mL, R^2^ = 0.9997			R^2^ > 0.99
Accuracy	16 µg/mL		98	1.074	<2
	20 µg/mL		99.680	0.933	<2
	24 µg/mL		99.020	0.343	<2
Interday precision	16 µg/mL		100.350	1.910	<2
	20 µg/mL		100.105	0.502	<2
	24 µg/mL		99.688	0.632	<2
Intraday precision 20 µg/mL			100.275	1.259	<2
Robustness in PBS, DW & SSF at 361 & 364 nm			99.274	1.786	<2
Method sensitivity	LOD	0.192 µg/mL			
	LOQ	0.582 µg/mL			
Stability in SSF at PH 6.8	1 week		100.690	1.898	
	3 months		96.456	0.342	

**Table 2 pharmaceuticals-16-01412-t002:** The result of the weight, hardness, and disintegration tests for the initial formulas of sublingual mucoadhesive tablets.

Formula	Weight (mg) ± SD *	Hardness ± SD *	Disintegration Time (min) ± SD *	Result
1	39.711 ± 3.667	Very friable	1.624 ± 0.266	Fail
2	49.387 ± 1.806	4.933 ± 0.404	0.796 ± 0.124	Pass
3	50.967 ± 0.603	6.367 ± 0.513	0.514 ± 0.0359	Pass
4	50.477 ± 1.581	3.867 ± 0.351	65.200 ± 4.850	Fail
5	50.043 ± 0.569	4.267 ± 0.322	64 ± 3.667	Fail
6	50.533 ± 1.234	4.700 ± 0.400	60.400 ± 0.693	Fail
7	52.367 ± 1.193	4.833 ± 0.567	65.540 ± 5.093	Fail
8	50.233 ± 0.851	5.067 ± 0.252	46.667 ± 2.880	Fail
9	51.633 ± 0.839	5.133 ± 0.351	50 ± 2	Fail
10	50.567 ± 1.531	5.867 ± 0.306	15 ± 1	Fail
11	50.667 ± 1.168	Very friable	15 ± 2	Fail
12	50.333 ± 1.289	Very friable	14.667 ± 2.517	Fail
13	49.467 ± 0.379	5.7 ± 0.436	55 ± 2	Fail
14	50.433 ± 1.002	3.833 ± 0.208	32.330 ± 2.517	Fail
15	50.100 ± 0.985	4.200 ± 0.300	40.167 ± 0.763	Fail
16	50.100 ± 1.039	5.267 ± 0.351	37 ± 2	Fail
17	50.200 ± 0.693	5.100 ± 0.265	10.007 ± 1.550	Fail
18	51.097 ± 0.700	4.167 ± 0.451	5 ± 1.500	Fail
19	50.833 ± 0.500	4.033 ± 0.153	0.234 ± 0.026	Pass
20	50.433 ± 2.201	5.233 ± 0.252	0.667 ± 0.009	Pass
21	49.833 ± 1.935	3.8 ± 0.200	0.501 ± 0.018	Pass
22	50.567 ± 1.504	5.333 ± 0.252	1.661 ± 0.168	Pass
23	50.500 ± 1.473	4.067 ± 0.503	13.710 ± 1.391	Fail
24	50.437 ± 0.753	5.100 ± 0.300	1.021 ± 0.188	Pass
25	49.700 ± 1.100	5.200 ± 0.265	0.846 ± 0.020	Pass
26	52.300 ± 0.755	6.533 ± 0.306	46.093 ± 2.220	Fail
27	50.533 ± 0.907	5.067 ± 0.252	0.667 ± 0.033	Pass
28	49.467 ± 0.850	6.033 ± 0.551	2.587 ± 0.161	Pass
29	51.200 ± 0.529	4.833 ± 0.208	0.192 ± 0.016	Pass
30	51.300 ± 0.819	3.670 ± 0.252	0.235 ± 0.009	Pass
31	50.567 ± 0.907	4 ± 0.200	0.260 ± 0.033	Pass
32	50.3 ± 1.769	5.933 ± 0.404	0.328 ± 0.016	Pass
33	50.367 ± 0.683	4.300 ± 0.400	0.276 ± 0.023	Pass
34	50.700 ± 1.400	3.733 ± 0.306	0.156 ± 0.012	Pass
35	50.407 ± 1.485	5.200 ± 0.265	0.194 ± 0.014	Pass
36	49.967 ± 2.759	5.167 ± 0.416	0.225 ± 0.034	Pass
37	51.267 ± 1.168	4.433 ± 0.503	0.266 ± 0.027	Pass
38	50.067 ± 1.041	4.800 ± 0.200	0.206 ± 0.027	Pass
39	51.600 ± 1.082	5.400 ± 0.460	0.441 ± 0.060	Pass
40	51.467 ± 1.069	5.267 ± 0.252	0.361 ± 0.035	Pass
41	52.067 ± 1.102	3.966 ± 0.351	0.245 ± 0.019	Pass
42	50.600 ± 0.400	5.100 ± 0.265	0.310 ± 0.033	Pass
43	51.533 ± 0.551	5.600 ± 0.400	0.264 ± 0.034	Pass

* Average of triplicate.

**Table 3 pharmaceuticals-16-01412-t003:** The result of the weight, hardness, and disintegration tests for the formulas of sublingual mucoadhesive cyanocobalamin tablets.

Formula	Polymers	Hardness ± SD *	Weight ± SD (mg) *	Disintegration Time ± SD (s) *	Residence Time ± SD (min) *	Result
S1	EC 5%	5.200 ± 0.600	49.600 ± 3.500	16.290 ± 4.050	1.240 ± 0.085	Fail
S2	EC 10%	4.200 ± 0.300	51.560 ± 1.300	24.590 ± 6.190	1.350 ± 0.280	Fail
S3	EC 15%	5.100 ± 0.900	51.230 ± 2.520	30.530 ± 8.840	2.190 ± 0.612	Fail
S4	HPMC 1%	4.500 ± 0.100	49.470 ± 0.450	24.370 ± 7.020	5.110 ± 2.150	Fail
S5	HPMC 5%	5.770 ± 0.378	49.170 ± 3.540	42.450 ± 16.430	86.400 ± 48.170	Pass
S6	HPC 5%	4.733 ± 0.321	53.200 ± 1.720	25.360 ± 2.250	4.200 ± 4.790	Fail
S7	HPC 10%	5.800 ± 0.608	51.830 ± 1.300	31.980 ± 3.960	3.550 ± 1.340	Fail
S8	HPC 15%	5.600 ± 0.400	49.630 ± 1.770	42.780 ± 9.340	22.450 ± 6.470	Pass
S9	EL100-55 5%	4.300 ± 0.608	50.100 ± 1.810	18.870 ± 2.310	5.260 ± 10.560	Fail
S10	EL100-55 10%	6 ± 0.600	50.630 ± 2.520	14.410 ± 3.070	4.160 ± 3.810	Fail
S11	EL100-55 15%	3.500 ± 0.458	50.130 ± 2.690	16.680 ± 1.750	118.200 ± 2.890	Pass
S12	EL100 5%	4.700 ± 0.700	50.970 ± 1.460	17.600 ± 3.250	0.742 ± 0.106	Fail
S13	EL100 10%	5.700 ± 0.435	51.330 ± 1.620	22.310 ± 1.500	1.470 ± 20.820	Fail
S14	EL100 15%	3.700 ± 0.435	52.500 ± 2.150	30.723 ± 9.830	5.660 ± 4.130	Fail
S15	ES100 5%	4.400 ± 0.624	53.200 ± 3.160	20.003 ± 1.370	2.180 ± 1.140	Fail
S16	ES100 10%	3.100 ± 0.100	51.267 ± 1.540	16.257 ± 1.470	1.410 ± 0.430	Fail
S17	ES100 15%	3.700 ± 0.556	52.100 ± 2.440	17.720 ± 0.540	4.450 ± 0.980	Fail
S18	CR490 0.5%	3.800 ± 0.608	52.733 ± 0.750	20.247 ± 0.690	5.330 ± 3.470	Fail
S19	CR490 1%	4.600 ± 0.458	50.733 ± 1.500	58.340 ± 1.480	4.650 ± 2.110	Fail
S20	XAN 0.5%	5.200 ± 0.100	51.300 ± 1.630	53.930 ± 2.370	57.400 ± 19.660	Pass

* Average of triplicate.

**Table 4 pharmaceuticals-16-01412-t004:** The evaluation of flow characteristics of the final formula blend.

Formula	Polymer	Angle of Repose ± SD *	Carr’s Index ± SD *	Bulk Density ± SD *	Tapped Density ± SD *	Hauser Test ± SD *
S5	HPMC	21.047 ± 2.931	15.789 ± 0.432	0.350 ± 0.075	0.415 ± 0.087	1.187 ± 0.006
S8	HPC	15.836 ± 4.866	12.766 ± 0.405	0.363 ± 0.050	0.416 ± 0.059	1.146 ± 0.005
S11	Eudragit L 100-55	23.400 ± 1.536	17.857 ± 6.409	0.355 ± 0.028	0.433 ± 0.001	1.270 ± 0.095
S20	Xanthan gum	23.505 ± 0.042	12.727 ± 1.889	0.362 ± 0.074	0.415 ± 0.093	1.146 ± 0.025

* Average of triplicate.

**Table 5 pharmaceuticals-16-01412-t005:** Physical properties of mucoadhesive cyanocobalamin sublingual tablets.

Formula	Weight Variation mg ± SD *	Diameter mm ± SD *	Thickness mm ± SD *	Hardness Kp ± SD *	Friability % ± SD *	Assay % ± SD *	Surface PH ± SD*	Mucoadhesive Strength (g) ± SD *
S5	50.320 ± 0.591	5 ± 0	1.5 ± 0	4.940 ± 0.742	0.480 ± 0.078	93.508 ± 0.001	6.630 ± 0.010	14 ± 1.732
S8	49.935 ± 0.668	5 ± 0	2 ± 0	4.280 ± 0.933	0.367 ± 1.245	103.910 ± 0.004	6.410 ± 0.020	11 ± 1
S11	50.155 ± 0.638	5 ± 0	2 ± 0	4.910 ± 0.935	0.304 ± 0.808	99.124 ± 0.002	5.350 ± 0.026	26 ± 1
S20	50.085 ± 0.774	5 ± 0	2 ± 0	4.609 ± 0.943	0.262 ± 0.060	96.362 ± 0.0008	6.490 ± 0.006	18.670 ± 1.528

* Average of triplicate.

**Table 6 pharmaceuticals-16-01412-t006:** Release kinetics results for final formulas for first 5.5 h (S5, S8, S11, and S20).

Model	Zero Order		First Order		Higuchi Model		Quadratic Model			Makoid-Banakar Model				Peppas–Sahlin Model			
Formula	K0	R2	K1	R2	KH	R2	K1	K2	R2	KMB	N	K	R2	K1	K2	m	R2
STD	16.001	0.7164	0.253	0.9122	28.542	0.9395	-0.025	0.256	0.9200	25.240	0.527	−0.031	0.9623	16.106	9.794	0.415	0.9818
S5 (HPMC)	4.941	0.9552	0.055	0.9366	9.292	0.7382	0.005	0.029	0.9829	2.429	2.290	0.261	0.9953	−20.494	21.193	0.327	0.9990
S8 (HPC)	4.964	0.9810	0.056	0.9729	9.497	0.8120	0.001	0.044	0.9834	3.858	1.812	0.207	0.9930	−39.880	41.925	0.202	0.9977
S11 (EL100-55)	7.584	0.9571	0.091	0.9324	14.357	0.7593	0.005	0.055	0.9694	54.574	2.238	0.289	0.9858	−77.813	78.491	0.190	0.9949
S20 (XG)	8.572	0.9638	0.108	0.9808	16.754	0.8931	−0.005	0.110	0.9822	10.370	1.638	0.251	0.9917	−1350.432	1356.751	0.015	0.9969

**Table 7 pharmaceuticals-16-01412-t007:** The result of the cyanocobalamin permeability test, including R^2^ for the Peppas–Sahlin model, flux steady state, and apparent permeability coefficients (*P_app_*).

Formula	R^2^	Flux (µg/h/cm^2^;) ± SD *	*P_app_* (cm/h) ± SD *	*P_app_* (cm/s)
S5 (HPMC)	0.9825	3.084 ± 0.376	0.0062 ± 0.0008	1.713 × 10^−6^
S8 (HPC)	0.9875	3.937 ± 0.240	0.0079 ± 0.0005	2.188 × 10^−6^
S11 (EL 100-55)	0.9987	6.387 ± 1.860	0.0127 ± 0.0037	3.548 × 10^−6^
S20 (XG)	0.9863	4.623 ± 1.322	0.0092 ± 0.0026	2.568 × 10^−6^

* Average of triplicate.

**Table 8 pharmaceuticals-16-01412-t008:** Result of cyanocobalamin stability in the simulated salivary fluid solution at 37 °C.

Time (h)	Concentration (µg/mL) ± SD *	% Recovery ± SD *	Drug Amount (µg) ± SD *	RSD of % Recovered
Zero	19.659 ± 0.084	100 ± 0	9829.65 ± 42.197	0
1	19.567 ± 0.084	99.537 ± 0.429	9783.61 ± 42.197	0.431
2	19.457 ± 0.287	98.957 ± 1.597	9820.442 ± 143.540	1.598
3	19.370 ± 0.032	98.530 ± 0.423	9673.112 ± 15.949	0.430
4	19.510 ± 0.084	99.240 ± 0.854	9755.985 ± 42.197	0.861
5	19.309 ± 0.084	99.920 ± 0.836	9663.904 ± 42.197	0.850
6	19.198 ± 0	99.350 ± 0.419	9599.448 ± 0	0.429
24	18.297 ± 0.055	94.660 ± 0.620	9157.459 ± 27.624	0.665

* Average of triplicate.

**Table 9 pharmaceuticals-16-01412-t009:** The result of cyanocobalamin stability after being stored in simulated saliva fluid.

At Zero Time		After 1 Week		After 3 Months	
Con (20 µg/mL)		Con (20 µg/mL)		Con (20 µg/mL)	
Mean abs ± SD	0.381 ± 0.006	Mean abs ± SD	0.380 ± 0.007	Mean abs ± SD	0.365 ± 0.001
RSD	1.461	RSD	1.898	RSD	0.342
SE	0.003	SE	0.004	SE	0.0007
CI	0.381 ± 0.006 0.375–0.387	CI	0.380 ± 0.008 0.372–0.388	CI	0.365 ± 0.001 0.364–0.366
Con found.	20.193	Con found.	20.138	Con found.	19.290
% Recovery	100.960	% Recovery	100.690	% Recovery	96.456

Measurement of triplicate, Con: concentration, abs: absorbance, RSD: relative standard deviation, SD: Standard deviation, CI: confidence interval.

## Data Availability

The data used to aid the outputs of this research are available from Hani Naseef (hshtaya@birzeit.edu) upon request.

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
