# Peer review of "The Preparation and Evaluation of Cyanocobalamin Mucoadhesive Sublingual Tablets"

_pharmaceuticals, 2023, doi:10.3390/ph16101412_

Round 1

Reviewer 1 Report

Dear authors, kindly find the following comments regarding your submitted manuscript:

1-     The overall idea of the research is good and useful.

2-     It is generally well-written but, it needs grammatical revision overall the manuscript.

3-     The abstract needs to be greatly modified. It should contain exact numbers and values of the measured evaluating parameters.

4-     The introduction is well-written.

5-     Materials and methods section should be written before the results and discussion section.

6-     Section “3.2” named instruments, I guess every instrument should be mentioned in its place inside the materials and methods section. It should not be written as a separate part.

7-     In table (3), standard deviations should be added and table (4) also, standard deviation values should be adjusted and missing ones should be added.

8-     In table (6), standard deviations for the flux values should be added.

9-      Also, in table (7), standard deviations for RSD of % recovered should be added.

English needs grammatical check.

Author Response

Dear Reviewer

We appreciate input. We carefully evaluated the valuable comments and adjusted the manuscript as necessary. Our point-by-point responses and the related adjustments are attached. A colleague from the USA also checked the manuscript for language.

The references were thoroughly checked to ensure that they were relevant to the manuscript's contents.

Please be aware that the Changes Outline report addresses the correction

Reviewer 2 Report

Dear authors

The manuscript is beautifully designed, a lot of effort and work was put into it, the formulations are innovative and  i hope will have an application in the future

Here are my questions for you

All formulations have a pH that questions stability B12. Optimal pH for B12 stability is between 4.5-5.5. Have you done forced degradation studies

Why the UV- VIS method was chosen for monitoring the dissolution rate. You say that it is validated according to ICH guidelines, but nowhere is there any data that indicates that, at least tabularly. If you use the method to determine the content according to the ICH guideline, there is no need to determine the LOQ and LOD

Why didn't you use some separation technique, such as HPLC? That's where you could see the behavior of your B12

No need to you burden your work with images of instruments

Author Response

(The authors gave the same response as above.)

Reviewer 3 Report

Authors of the present manuscript reports results of development of a sublingual tablet’s formulation with cyanocobalamin as an API. The presented study is experimentally extensive but of very low scientific value and so not appropriate for publishing in high ranked scientific journal.

What is the reason for the authors’ decision to develop prolonged release formulations of cyanocobalamin?

Authors should very also chemical stability of the API in the optimal formulations through  analysing the content and profile od API’s related substances. The same should be used in evaluation of API stability in simulated saliva.

There are many terminology and language weaknesses in the manuscript, which should be improved.

Author Response

(The authors gave the same response as above.)

Reviewer 4 Report

The significant figures/digits (number of digits in a value, that contribute to the degree of accuracy of the value) must be rationalized in each table. In many cases these numbers are chaotic.  

Figures 5 -8 are not necessary for the manuscript, perhaps they can appear as supporting material, but there is no real need for this either.

The evaluation of the dissolution measurements must be repeated. The kinetic models must be fitted to the data recorded in the first 5 hours of the measurement, so it is highly likely that completely different fitting results will be obtained. In Figure 3, in addition to the data measured in the five hours periods, the calculated curves of the best-fitting models should also be shown.

Quality of Figures 3 and 4 definitely need to be improved, as they are currently the only edited graphs in the manuscript.

The R2 values ​​of Table 5. suggest that a linearized form of the kinetic models was used. This is completely unnecessary because the nonlinear evaluation can be easily performed using any spreadsheet software. (Aeasy-to-follow example can be seen here: https://www.youtube.com/watch?v=rln0IqI50Iw) If they don't want to spend time on it, Excel table for nonlinear evaluation can be freely downloaded and used from the supporting material of the article below.

https://doi.org/10.1016/j.molliq.2021.115405

Author Response

(The authors gave the same response as above.)

Round 2

Reviewer 2 Report

The answers you gave to my questions are not complete and well-argued.

Reviewer 3 Report

The Authors made some corrections of the manuscript, which on my opinion didn’t improve the overall scientific value of the manuscript. Some open remarks:

1.       The manuscript lack “in vivo” verification of the optimal formulations as it is difficult to explain how this formulations e.g. S11 would behave in “in vivo” conditions in respect of the give results from table 5 (disintegration time – approx. 16 seconds and residence time approx. 118 minutes) and Figure 4 – less than 80% of API was released in 24 hour. These facts should be scientifically sound explained by the autors.

2.       Authors should explain why only first the release profile in the 5.5 hours timeframe was used in elaboration of drug release kinetic mechanism in which only less than 50% of the incorporated drug was released.

3.       The authors should scientifically sound discuss the stability of the formulation (at least optimal) per see – stability of the API, which is chemically highly sensitive, in tablets on stress conditions.  The stability study presented in Table 11 – stability of the API in water for 1 week or 3 months is not relevant nor for the formulation nor for the API.
